# Overview on the Development of Alkaline-Phosphatase-Linked Optical Immunoassays

**DOI:** 10.3390/molecules28186565

**Published:** 2023-09-11

**Authors:** Lin Liu, Yong Chang, Jiaxin Lou, Shuo Zhang, Xinyao Yi

**Affiliations:** 1College of Chemistry and Chemical Engineering, Anyang Normal University, Anyang 455000, China; 2College of Chemistry and Chemical Engineering, Central South University, Changsha 410083, China

**Keywords:** alkaline phosphatase, immunoassays, fluorescence, colorimetry, chemiluminescence, surface-enhanced Raman scattering

## Abstract

The drive to achieve ultrasensitive target detection with exceptional efficiency and accuracy requires the advancement of immunoassays. Optical immunoassays have demonstrated significant potential in clinical diagnosis, food safety, environmental protection, and other fields. Through the innovative and feasible combination of enzyme catalysis and optical immunoassays, notable progress has been made in enhancing analytical performances. Among the kinds of reporter enzymes, alkaline phosphatase (ALP) stands out due to its high catalytic activity, elevated turnover number, and broad substrate specificity, rendering it an excellent candidate for the development of various immunoassays. This review provides a systematic evaluation of the advancements in optical immunoassays by employing ALP as the signal label, encompassing fluorescence, colorimetry, chemiluminescence, and surface-enhanced Raman scattering. Particular emphasis is placed on the fundamental signal amplification strategies employed in ALP-linked immunoassays. Furthermore, this work briefly discusses the proposed solutions and challenges that need to be addressed to further enhance the performances of ALP-linked immunoassays.

## 1. Introduction

Immunoassays have become the most widely utilized detection techniques in various fields, such as food safety, disease diagnosis, and environmental monitoring [1,2,3]. The demand for on-site and real-time detection of disease biomarkers has led to the emergence of optical immunoassays as effective technologies in biological analysis [4,5,6,7]. With a physical optical transducer, the interactions between antibodies and antigens can be translated into detectable optical signals that change linearly with the target concentration. Currently, well-developed optical immunoassays are primarily classified into fluorescence, colorimetry, chemiluminescence, and surface-enhanced Raman scattering (SERS) based on their signal detection formats [8,9,10,11]. In comparison to electrochemical immunosensors, optical immunoassays offer significant advantages, such as low sample matrix effect, straight forward operation procedure, high signal-to-noise ratio, and wide applicability for on-site detection of diverse targets. To enhance the sensitivity in determining ultralow-abundance targets, several effective signal amplification strategies have been integrated with optical immunoassays, including enzyme catalysis, DNA-based amplification, and nanotechnology [12,13,14,15,16]. For instance, semiconductor quantum dots (QDs), carbon dots (CDs), and noble metal nanoclusters have been widely employed as novel probes with attractive photoluminescence features, replacing conventional fluorescent dyes [17,18,19]. Gold nanoparticles (AuNPs) can serve as carriers for loading numerous enzymes to achieve multiple-signal amplification, and as chromogenic indicators with high extinction coefficients as well as size- and inter-particle-distance-dependent optical properties [20,21,22,23,24]. However, despite their widespread application, some of these amplification strategies still suffer from drawbacks such as complex synthesis processes and high toxicities of heavy metal ions. Enzyme-linked immunoassays, integrating the inherent sensitivity and specificity of immune reactions and the high efficiency of enzymatic catalysis, have gained popularity in bioanalytical fields [25].

Nowadays, different natural catalytic enzymes, including horseradish peroxidase (HRP), alkaline phosphatase (ALP), glucose oxidase (GOx), and catalase (CAT), have been successfully introduced into immunoassays as labels to generate detectable signals [26,27,28]. Among them, HRP and ALP are the two most popularly used reporter enzymes for high-performance bioanalysis [29]. For example, the commercially available enzyme-linked immunosorbent assays (ELISAs) with antibody or antigen-conjugated HRP can catalyze the oxidation of chromogenic substrates into colored molecules using H_2_O_2_. Nonetheless, the use of HRP is always affected by several inherent issues, such as a non-specific staining response, activity inhibition by Cu^+^ ions, various micro-organisms and antibiotics, as well as the requirement of an unstable H_2_O_2_ and carcinogenic substrate for enzymatic reactions [30,31,32,33,34]. In contrast, ALP has garnered considerable attention as a reporter enzyme for signal amplification due to its outstanding advantages of high catalytic activity, high turnover number, and broad substrate specificity [35,36,37]. Several review papers have summarized the methods for ALP activity assays [18,38,39,40].

Despite the high catalytic activity and specificity of ALP, the moderate sensitivity of enzyme-linked immunoassays may restrict practical applications for determining low-abundance targets [37,41]. To achieve a higher sensitivity and lower limit of detection (LOD), various strategies and devices have been meticulously combined with ALP-based signal amplification to boost the performances of immunoassays. Examples include nanomaterials modified with multiple ALP molecules and cascade reactions between ALP and nanocatalysts/nanozymes to construct immunoassays with diverse schemes and protocols [42,43]. Additionally, converting immune reaction events into DNA amplification reactions and utilizing DNA nanostructures to capture a large number of ALP molecules have enabled multiple-signal amplification [13]. Notably, the combination of ALP catalysis and plasmonic colorimetric reactions has opened up possibilities for developing immunoassays with multi-color changes and the applicability to point-of-care testing (POCT) systems [44,45,46]. Numerous ALP-linked immunoassays have been reported in recent years, and some reviews have briefly mentioned the roles of ALP in detection principles [40,47,48,49]. However, few reviews have specifically focused on the advancements in optical immunoassays using ALP as the signal label. Thus, this comprehensive review aims to summarize the progress in ALP-linked optical immunoassays, covering fluorescence, colorimetry, chemiluminescence, and SERS. Each section categorizes immunoassays carefully based on the role of ALP and the enzymatic product, providing valuable insights for potential researchers.

## 2. Fluorescence Immunoassays

In ALP-linked fluorescence immunoassays, a broad variety of fluorescent materials have been introduced, including organic dyes, QDs, metal nanoclusters (NCs), CDs, and metal–organic frameworks (MOFs) [50,51,52]. Theoretically, the previously reported fluorescence strategies for ALP-targeting assays can be designated to quantify various antigens in the form of immunoassays using ALP as the reporter enzyme [17,18,39,53]. This section categorizes the ALP-linked fluorescence immunoassays according to the signal output methods (Figure 1): the direct generation of fluorescent molecules or quenchers, ALP catalysis coupling with chemical reaction or enzymatic cascade reaction, enzymatic-product-regulated fluorescence of nanomaterials, enzymatic-product-induced in situ generation of fluorescent nanomaterials, and enzymatic-product-triggered aggregation-induced emission (AIE) phenomenon (Table 1).

### 2.1. Direct Generation of Fluorescent Molecules or Quenchers

Among the kinds of ALP-linked fluorescence immunoassays, it is the simplest strategy for target detection to compare the fluorescence signal of the substrate and enzymatic product (Figure 2) [54,55,56,57]. Liu et al. developed a direct competitive fluorescence immunoassay for ochratoxin A (OTA) detection based on ALP catalysis [58]. In this study, a nanobody-alkaline phosphatase (ALP) fusion protein was directly expressed in bacteria through antibody engineering and the molecular cloning technique. ALP catalyzed the hydrolysis of the weak fluorescent 2′-(2-benzothiazoyl)-6′-hydroxybenzothiazole phosphate into a strong fluorescent 2′-(2-benzothiazoyl)-6′-hydroxybenzothiazole (Figure 2). Based on the difference in the fluorescence of the substrate and product, OTA was determined with an LOD of 0.04 ng/mL and a linear range of 0.06–0.43 ng/mL. Furthermore, for the development of the automated, sensitive, and straightforward diagnosis of toxoplasmosis, Medawar-Aguilar et al. constructed a microfluidic-laser-induced fluorescence immunosensor for determining anti-Toxoplasma gondii immunoglobulin G (IgG) (anti-*T. gondii*)-specific antibodies (Figure 1) [59]. Chitosan-ZnO-nanoparticles were employed to immobilize *T*. *gondii* antigens into the central microfluidic channel. ALP-labeled anti-IgG antibodies were used to recognize the captured antibodies. ALP catalysis promoted the transformation of non-fluorescent 4-methylumbelliferylphosphate (4-MUP) into soluble fluorescent methylumbelliferone (4-MU). The fluorescence intensity at 440 nm was proportional to the level of anti-*T. gondii*-specific antibodies. To achieve the single-molecule detection sensitivity, ALP-based enzymatic catalysis can be introduced into digital immunoassays [26,60,61]. For example, Tsaloglou et al. reported a magnetically assisted fluorogenic heterogeneous immunoassay of cardiac marker Troponin I (cTnI) on a low-voltage digital microfluidic platform [62]. A narrow bridge was implemented on the platform to facilitate the washing procedure, leading to more than 90% of removal of unbound reagents in five washes. In this work, 9*H*-(1,3-dichloro-9,9-dimethylacridin-2-one-7-yl) (DADO) phosphate was employed as the fluorogenic substrate.

ALP-catalytic product *p*-nitrophenol (PNP) with an absorption band at 400 nm in the anionic form can quench the fluorescence of dyes via the photoinduced electron transfer (PET) or Fӧrster resonance energy transfer (FRET) mechanism [63]. Therefore, the fluorescence “turn-off” detection mode can be combined with ALP-linked immunoassays based on the quenching ability of enzymatic products. Recently, Ma et al. developed a fluorescence and colorimetric dual-mode immunoassay for the detection of zearalenone (ZEN) based on G-quadruplex/N-methylmesoporphyrin IX (NMM) [64]. As displayed in Figure 2, G-quadruplex (G4) was formed through the hydrogen bond interactions between four guanines in G-rich DNA and NMM to generate a strong fluorescence signal. ALP-catalyzed production of yellow PNP effectively quenched the fluorescence of G4/NMM. ZEN was determined by monitoring the visual color of PNP and the fluorescence of G4/NMM.

### 2.2. Generation of Fluorescent Molecules through Chemical Reaction or Enzymatic Cascade Reaction

Compared with ascorbic acid 2-phosphate (AAP), the enzymatic product ascorbic acid (AA) in the dehydrogenized format can quickly react with *o*-phenylenediamine (OPD) to form *N*-heterocyclic fluorophore under alkaline conditions (Figure 2) [65,66]. For this view, Zhao et al. developed an immunosensor for *α*-fetoprotein (AFP) detection through the ALP-triggered in situ generation of fluorescent molecules [67]. As displayed in Figure 3A, the enzymatic product AA reacted with OPD to form 3-(1,2-dihydroxyethyl)furo[3,4-b]quinoxalin-1(3H)-one, which emitted a blue fluorescence. The extent of fluorescence intensity was directly associated with the concentration of AA, which was dependent upon the enzymatic reaction and AFP content. The group further designed an original substrate *m*-hydroxyphenyl phosphate sodium salt for ALP catalysis. The enzymatic product resorcinol could interact with dopamine through nucleophilic reaction, producing fluorometric and colorimetric dual signals [68]. In addition, Fan et al. reported an immunoassay for carcinoembryonic antigen (CEA) detection based on the reaction between AA and terephthalic acid (PTA) (Figure 3B) [69]. In this study, AA promoted the generation of hydroxyl radicals (^•^OH) in the presence of O_2_ under high temperature. The produced ^•^OH could react with PTA to form a blue-fluorescent product of PTA-OH. By coupling this reaction with ALP-linked immunoreaction, CEA was sensitively detected in the range of 0.25–30 ng/mL with an LOD of 0.08 ng/mL.

Many ALP-linked fluorescence immunoassays have been developed based on the difference in the binding affinity between enzyme substrates and products toward substances [70]. Recently, Geng et al. reported an ALP-linked fluorescence immunosensor for the detection of SARS-CoV-2 N protein and cTnI [71]. In this work, the self-assembles formed between pyridineboronic acid (PyB(OH)_2_) and alizarin red S (ARS) showed a strong fluorescence. However, PPi could preferentially interact with PyB(OH)_2_, leading to the quenching of fluorescence. When PPi was catalytically decomposed into Pi ions, PyB(OH)_2_ was reacted with ARS to recover the fluorescence.

In addition, an ALP substrate can be partially conjugated with a fluorescent moiety, and its coordination interaction with metal ions can be modulated through ALP-catalyzed dephosphorylation. Lin et al. reported an ALP-linked fluorescence immunoassay based on the PET process between Fe(III) and the ATP-BODIPY conjugate [72]. As shown in Figure 4A, ATP-BODIPY could bind to Fe(III) and its fluorescence was quenched by Fe(III) through the PET mechanism. The ATP-BODIPY conjugate could be enzymatically dephosphorylated into BODIPY-adenosine that was insensitive to Fe(III), thereby restoring the fluorescence of BODIPY. Moreover, the fluorescence of fluorochrome can be quenched by metal ions and ALP-catalyzed products (inorganic phosphate ions) can competitively bind with metal ions to retrieve the fluorescence [73]. Chen et al. developed a fluorescence immunoassay based on Pi-triggered fluorescence “turn-on” determination of AFP (Figure 4B) [74]. In this study, the yellow green fluorescence of calcein was dramatically quenched by Ce^3+^ via the specific coordination interaction. After the immunoreaction, Pi, produced from the ALP-catalyzed hydrolysis of substrate *p*-nitrophenyl phosphate (PNPP), was competitively chelated with Ce^3+^ to recover the fluorescence of calcein.

The signal generated by a single enzyme-linked reaction is relatively limited. Therefore, the combination of ALP-mediated enzymatic catalysis with other auxiliary amplification strategies has been proposed as a promising approach to improve the detection sensitivity and efficiency, such as enzyme cascade reactions, DNA-based amplification, and magnetic particles. Notably, enzymatic cascade reactions based on two natural enzymes have been constructed in different biosensing fields, in which the products from the first enzyme-linked catalysis can be directly used as the substrates of the second enzyme-involved reaction. Based on the classic system of ALP and tyrosinase (Tyr), Zhao et al. developed a dual-mode enzyme-cascade-based immunosensor for the detection of cTnI in diluted serums [75]. As shown in Figure 5, after the formation of the immune-complex, ALP catalyzed the hydrolysis of PAPP into the intermediate tyramine that could be hydroxylated into dopamine through Tyr catalysis. The generated dopamine was further reacted with resorcinol to produce azamonardine. The resulting solution exhibited a pale yellow color and emitted intense blue fluorescence. Finally, 0.015 ng/mL or 0.06 ng/mL of cTnI could be sensitively detected with a fluorescence spectrometer or a UV spectrometer, respectively. The enzymatic cascade reactions can significantly amplify the signals. However, the suitable ALP-based enzyme pairs are still very limited.

Although natural enzymes exhibit high substrate specificity and catalytic efficiency, they face several inherent drawbacks, such as a time-consuming and expensive preparation, poor stability, and inferior tolerance to harsh environmental conditions. Therefore, organic molecules, metal complexes, and nanomaterials with enzyme-like characteristics have been prepared to replace or couple with natural enzymes in chemical and biochemical assays [76,77,78]. For example, Li et al. developed a ratiometric fluorescence immunoassay based on the competitive consumption of OPD between ALP catalysis and nanozyme catalysis (Figure 6) [79]. In this work, catechol-oxidase-like nanozyme (CHzyme) was prepared to catalyze the conversion of the phenolic hydroxyl group of catechol into a carbonyl group. CHzyme-catalyzed product *o*-benzoquinone could further react with OPD via the Schiff-based chemistry to generatea fluorescent substance with an emission at 560 nm. However, the ALP-catalyzed product AA could interact with OPD to form fluorescent species with an emission at 425 nm. The competitive interaction of ALP-catalyzed and CHzyme-based products led to the increase in emission intensity at 425 nm and the decrease in emission intensity at 560 nm. The ALP-based immunoassay achieved the sensitive detection of clenbuterol with an LOD of 0.017 ng/mL based on the ratiometric fluorescence sensing mode.

### 2.3. Enzymatic-Product-Regulated Fluorescence of Nanomaterials

Fluorescence dyes may suffer from the limitations of pH sensitivity, hydrophobicity, and photobleaching. In order to effectively address these disadvantages, nanomaterials have been exploited as alternative fluorescent substrates in immunoassays, particularly including CDs, AuNCs, and QDs. These fluorescent nanomaterials possess versatile advantages, such as high fluorescence quantum yield, excellent photochemical stability, and tunable excitation and emission wavelength. ALP-based enzymatic substrates or products can regulate the fluorescence of nanomaterials, thereby allowing for the construction of “turn-on” or “turn-off” immunoassays.

The phosphate matrix for ALP can be used as an organic ligand to fabricate an infinite coordination polymer (ICP). The ALP-catalyzed hydrolysis of the ligand will induce the destruction of the ICP structure accompanied by the change in optical properties [80,81]. Li et al. reported a fluorescent immunoassay for mouse IgG detection based on ALP-responsive dye-doped ICP [82]. In this work, ICP composed of europium ions (Eu^3+^) and guanine monophosphate (GMP) was used to encapsulate fluorescent thioflavin T (ThT) molecules. The conformational rotation of ThT was restricted due to the confined effect of ICP, leading to the enhancement of ThT fluorescence. However, removing the phosphate group of GMP using ALP could destroy the structure of GMP/Eu ICPs, resulting in the release of ThT from the ICP composite and the decrease in fluorescence signal.

ALP-catalyzed products can serve as quenchers to decrease the fluorescence of nanomaterials in a “turn-off” detection approach [83,84,85]. In view of the fluorescence-quenching ability of ALP-enzymatic products, Li et al. developed an immunosensor for the detection of aflatoxin M_1_ residues in milk (Figure 7A) [86]. In this study, N-doped CDs with a quantum yield of 97.1% were prepared with citric acid and ethylenediamine as the precursors. During the label-free competitive immunoreaction, ALP immobilized on the plate catalyzed the hydrolysis of PNPP into PNP. The absorption spectrum of the product PNP overlapped with the emission spectrum of N-doped CDs, quenching the fluorescence through the inner filter effect (IFE). In addition, ALP-enzymatic products can react with other substances to produce quenchers, thus modulating the fluorescence of nanomaterials [87]. For example, the enzymatic product AA can reduce Au^3+^ and Ag^+^ into AuNPs and AgNPs to quench the fluorescence of CdSe/ZnS QDs and graphene quantum dots (GQDs), respectively [88,89]. Zhu et al. reported a fluorescence immunoassay for the detection of sulfamethazine based on glutathione (GSH)-capped silver nanoclusters (AgNCs) and ALP [90]. As shown in Figure 7B, ALP-labeled Ab_2_ was used to construct the competitive immunoassay. I_2_ was reduced into I^−^ by the product AA, quenching the fluorescence of AgNCs in an isopropanol buffer.

The above-mentioned “turn-off” immunoassays are always less sensitive due to their high background signals. Thus, a series of fluorescent “turn-on” immunoassays have been developed, in which the fluorescence of nanomaterials was quenched and then recovered by the products or by-products of ALP-mediated biocatalytic reactions through different mechanisms [91]. Metal ions, such as Ag^+^, Cu^2+^, and Fe^3+^, can be reduced by ALP-enzymatic products, modulating the fluorescence of nanomaterials [92,93]. Song et al. demonstrated that AA could reduce Fe^3+^ into Fe^2+^, restoring the fluorescence of CDs quenched by Fe^3+^. Then, they developed a “switch-on” fluorescent immunosensor for human IgG detection [94]. Zhou et al. developed a fluorescence immunoassay for the determination of ethyl carbamate based on magnetic particles and ALP catalysis [95]. In this work, the ALP-enzymatic AA could reduce Cu^2+^ into Cu^+^ that significantly quenched the fluorescence of CdSe QDs through a cation exchange mechanism. AA can also react with metal ions in MOFs and alter the optical and enzyme-like properties. Xie et al. reported a dual-mode fluorescent and colorimetric immunoassay for prostate-specific antigen (PSA) detection based on the AA-induced in situ generation of signals from MOFs [96]. As shown in Figure 8, Fe(III)-containing MOFs (Fe-MOFs) showing an oxidase-like activity and AA-responsive fluorescence emission were used as the multifunctional probes. In the sandwich immunoassay, ALP catalyzed the hydrolysis of AAP and the generated AA could reduce Fe(III) into Fe(II). The ligand-metal charge transfer (LMCT) from the fluorescent ligand 2-amino-1, 4-benzenedicarboxylic acid (BDC-NH_2_) to Fe(III) was blocked and the fluorescence of BDC-NH_2_ was restored, leading to the fluorescence recovery of Fe-MOFs. Meanwhile, the oxidase-like activity of Fe-MOFs decreased, inhibiting the oxidation of 3,3′,5,5′-tetramethylbenzidine (TMB) and the color change.

Fluorescent nanomaterials can be oxidized by chemical reagents along with the decrease in fluorescence intensity. However, the reduction of oxidized fluorescent nanomaterials by the ALP-enzymatic product may cause the fluorescence restoration. Based on this concept, Hu et al. developed a “switch-on” fluorescent immunosensor for mouse IgG detection on the basic of ALP and AuNCs [97]. As shown in Figure 9A, CaCO_3_–AuNPs were used to load the secondary antibody (Ab_2_) and signal reporter ALP. After the immunoreaction, AA produced from the ALP-catalyzed hydrolysis of AAP restored the fluorescence of AuNCs that was quenched by KMnO_4_. The fluorescence properties of CDs are closely related to the functional groups on their surface. Thus, the fluorescence can be adjusted by the oxidative and reductive agents via chemical reactions. Fang et al. developed a fluorescence immunoassay of AFP by tuning the surface state of CDs to modulate the fluorescence [98]. As illustrated in Figure 9B, AuNPs were simultaneously modified with Ab_2_ and capture probe DNA. Then, the hybridization chain reaction (HCR) occurred on the surface of AuNPs in the presence of biotin-labeled hairpin units, and more streptavidin (SA)-ALP conjugates were attached to the formed DNA polymers via the biotin-SA interactions. Under the ALP catalysis, the generated AA reduced the hydroxyl groups of CDs that were pre-oxidized by KMnO_4_, resulting in the recovery of fluorescence.

Nowadays, various FRET systems have been developed based on different nanomaterials. ALP-enzymatic products can regulate the FRET efficiency by destroying the structure of quenchers, thereby turning on the fluorescence. For example, V_2_O_5_ nanobelts, cobalt oxyhydroxide (CoOOH), and MnO_2_ NSs with a broad and intense adsorption spectrum can serve as efficient nanoquenchers to almost completely quench the fluorescence of different nanomaterials, such as CDs, upconversion nanoparticles (UCNPs), polydopamine, and GQDs [99,100,101,102]. The generated AA serving as a reducing agent can react with these nanomaterials to release metal ions and restore the fluorescence [103,104,105,106,107]. Li et al. developed a fluorescence immunoassay for imidacloprid detection by degrading the structure of CoOOH NSs by ALP-enzymatic products (Figure 10A) [108]. In this study, the negatively charged AuNCs were anchored on the surface of positively charged CoOOH NSs through the electrostatic interactions and the fluorescence was quenched via the FRET mechanism. After the competitive immunoreaction, ALP-labeled antibodies were tethered to the captured antigens. Under the ALP catalysis, the produced AA reduced the nanoquencher CoOOH NSs into Co^2+^ ions and the fluorescence of AuNCs was recovered. The CDs and MnO_2_ NS-based FRET system can also be coupled to ALP-linked immunoassays [109]. For example, Dong et al. developed a fluorescence immunoassay for the detection of amantadine with the assemblies of CDs and MnO_2_ NSs as the AA-responsive signal probes [110]. As shown in Figure 10B, CDs were immobilized on the surface of MnO_2_ NSs to form the FRET system. After the competitive immunoreaction, ALP in the immunocomplex catalyzed the production of AA that could reduce MnO_2_ NSs into Mn^2+^, recovering the fluorescence of CDs.

### 2.4. Enzymatic-Product-Induced In Situ Generation of Fluorescent Nanomaterials

It is attractive to develop fluorescent immunoassays based on the in situ growth of fluorescent nanomaterials triggered by the ALP-based biocatalytic process. The enzymatic product AA can reduce metal ions into metal nanoclusters in the presence of templates. Li et al. developed an ALP-linked fluorescent immunoassay for IgG detection based on the AA-reduced in situ generation of Cu nanoclusters (CuNCs) in the presence of dsDNA templates (Figure 11A) [111]. In this work, a sandwich immune system was constructed using ALP-modified secondary antibodies. ALP catalyzed the dephosphorylation of AAP to yield AA that could reduce Cu^2+^ ions in the presence of dsDNA to produce fluorescent CuNCs for IgG detection. Meanwhile, Yang’s group suggested that ALP-enzymatic product 4-aminophenol (AP) could react with *N*-[3-(trimethoxysilyl)propyl]ethylenediamine and ethylenediamine to form fluorescent silicon-containing nanoparticles and polymer CDs, respectively [112,113]. Sun et al. developed a fluorescence immunoassay for the detection of alpha-fetoprotein and human IgG via ALP-enabled in situ synthesis of fluorescent silicon nanoparticles (SiNPs) [114]. As shown in Figure 12, ALP catalyzed the transformation of AAP into AA that could further react with (3-aminopropyl) trimethoxysilane (APTMS) to form cyan fluorescent SiNPs.

Inorganic phosphate produced from the hydrolysis of substrates can be utilized to produce fluorescent nanomaterials. Malashikhina et al. developed a fluorescent immunoassay for anti-bovine serum albumin (anti-BSA) detection based on enzymatic formation of QDs (Figure 12) [115]. In this work, ALP catalyzed the hydrolysis of PNPP into PNP and phosphate (Pi) ions. After the addition of Cd^2+^ and S^2−^ ions, fluorescent CdS QDs were produced with Pi ions as the capping reagents and the signal was monitored by monitoring the emission spectra at *λ*_ex_ = 290 nm.

### 2.5. Enzymatic-Product-Triggered AIE Phenomenon

Since the discovery of the AIE phenomenon, AIE-based methods have opened up a field of analysis with significant potential applications [116,117,118]. The aggregation of fluorophores with AIE characteristics (AIEgens) results in a significantly enhanced and stable emission. Notably, the AIEgens-based signal output strategy has paved a new way for fluorescent immunoassays [119]. In particular, the ALP/AIEgens-based immunoassays integrate the advantages of the immune reaction, ALP catalysis, and AIE phenomenon, in which ALP catalysis can finely regulate the aggregation or disaggregation of AIEgens. Based on the hydrolysis of PPi into Pi, ALP-triggered “turn-off” AIE was employed by Liu and co-workers for the immune-sensing of IgG (Figure 13A) [120]. In this work, PPi was reacted with the tetraphenylethene (TPE) derivative with two diethylenetriamine (TPDA) groups through hydrogen linkage between the primary amine group and PPi. The PPi-induced self-assembly of TPDA activated the AIE effect, producing a strong fluorescence. During the ALP-labeled immunoreaction, PPi was hydrolyzed into Pi and the PPi-aided self-assembly of TPDA was inhibited, limiting the increase in fluorescence intensity. Cu(I)-catalyzed azide/alkyne cycloaddition (CuAAC) reactions have been widely used in bioconjugation and biosensing due to the mild reaction condition and wide applicability. Yuan et al. designed a self-clickable AIEgen-based signal amplification strategy for ALP-linked immunoassays [121]. In this work, the iconic AIEgen with TPE as the core was modified with two alkyne and two azide groups. The generated AA could reduce Cu^2+^ into Cu^+^, initiating the CuAAC reaction to form aggregates and thus lighting up the fluorescence. Recently, a broad variety of nanomaterials with AIE characteristics have been exploited to design fluorescent immunoassays. Chen et al. reported a Ce^4+^/Ce^3+^-triggered dual-readout immunoassay for OTA detection based on the AIE effect and TMB oxidation [122]. As presented in Figure 13B, ALP catalyzed the generation of AA from the substrate AAP. AA reduced Ce^4+^ into Ce^3+^ that could induce the AIE of AuNCs to enhance the fluorescence. Meanwhile, the unreacted Ce^4+^ ions could oxidize TMB into blue oxTMB.

As one of the most popular detection methods, the ALP-based fluorescence immunoassay shows high sensitivity, fast response, simple operation, and excellent stability. The emergence of novel fluorescence nanomaterials can overcome the defects of traditional fluorescent dyes in bioassays, such as low quantum efficiency, easy photobleaching degradation, and relatively short fluorescence lifetime. However, such fluorescence immunoassays still suffer considerable challenges, such as the susceptibility to matrix interference and the short lifespan of fluorophores. More efforts should be devoted to develop novel fluorescence materials with a long fluorescence lifetime and near-infrared emission wavelength and combine fluorescence technologies with integrated portable devices for real-time and rapid detection.

**Table 1 molecules-28-06565-t001:** Analytical performance of ALP-based fluorescence immunoassays.

Detection Principle	ALP Substrates	Fluorescence Reporters	Target	Linear Range	LOD	Reference
Direct generation of fluorescent molecules or quenchers	DDAO phosphate	DDAO	C-reactive protein	0.1–1000 ng/mL	58 pg/mL	[54]
	DDAO phosphate	DDAO	AIV H5-HA	0.23–100 ng/mL	0.23 ng/mL	[57]
	4-MUP	4-MU	Anti-*T-gondii* IgG antibodies	0–200 U/mL	0.39 mU/mL	[59]
	PNPP	G4/NMM	Zearalenone	7.5–17.5 ng/mL	36 pg/mL	[64]
Generation of fluorescent molecules through chemical reaction or enzymatic cascade reaction	AAP	*N*-heterocyclic fluorophore	AFP	0.5–40 ng/mL	0.21 ng/mL	[67]
	*m*-HPP	Azamonardine	cTnI	0.125–8 ng/mL	40 pg/mL	[68]
	AAP	PTA-OH	CEA	0.25–30 ng/mL	0.08 ng/mL	[69]
	BODIPY-ATP	BODIPY	IgG	0–200 ng/mL	5 ng/mL	[72]
	PNPP	Calcein	AFP	0.2–1 ng/mL	41 pg/mL	[74]
	PAPP	Azamonardine	cTnI	0.05–4 ng/mL	15 pg/mL	[75]
Enzymatic-product-regulated fluorescence of nanomaterials	GMP	ThT@GMP/Eu	Mouse IgG	0.8–100 ng/mL	0.16 ng/mL	[82]
	PNPP	CDs	Aflatoxin M_1_	0.003–0.81 ng/mL	18.6 pg/mL	[86]
	AAP	AuNCs	*Escherichia coli* O157:H7	3.3 × 10^3^–3.3 × 10^6^ cfu/mL	920 cfu/mL	[87]
	AAP	CdTe QDs	HIV-1 p24 antigen	1–100 pg/mL	0.2 pg/mL	[93]
	AAP	CDs	Human IgG	40 ng/mL–4 μg/mL	150 pg/mL	[94]
	AAP	CdSe QDs	Ethyl carbamate	100 ng/mL–10 μg/mL	24.3 ng/mL	[95]
	AAP	AuNCs	Mouse IgG	0.005–50 ng/mL	1.5 pg/mL	[97]
	AAP	CDs	Aflatoxin B1	1 ng/kg–1 μg/kg	0.69 ng/kg	[109]
Enzymatic-product-induced in situ generation of fluorescent nanomaterials	PAPP	Si CNPs	PSA	0.02–20 ng/mL	4.1 pg/mL	[112]
	PNPP	CdS QDs	Anti-BSA Antibody	0–500 ng/mL	0.4 ng/mL	[115]
Enzymatic-product-triggered AIE phenomenon	AAP	Self-clickable TPE-based AIEgens	Rabbit anti-human IgG	0–50 ng/mL	1.2 ng/mL	[121]
	AAP	AuNCs	Ochratoxin A	0–500 ng/mL	0.62 ng/mL	[122]

Abbreviation: DDAO phosphate, 9*H*-(1,3-dichloro-9,9-dimethylacridin-2-one-7-yl) phosphate; 4-MUP, 4-methylumbelliferyl phosphate; 4-MU,methylumbelliferone; AIV H5-HA, avian influenza virus H5-hemagglutinin; PNPP, *p*-nitrophenyl phosphate; G4/NMM, *G*-quadruplex/*N*-methylmesoporphyrin IX; CEA, carcinoembryonic antigen; AFP, α-fetoprotein; AAP, L-ascorbic acid 2-phosphate trisodium salt; PTA-OH, 2-hydroxyterephthalic acid; PAPP, *p*-aminoethyl-phenyl phosphate disodium salt; cTnI, cardiac troponin I; *m*-HPP, *m*-hydroxyphenyl phosphate sodium salt; BODIPY, boron dipyrromethene; BODIPY-ATP, BODIPY-conjugated adenosine triphosphate; IgG, immunoglobulin G; Si CNPs, silicon-containing nanoparticles; PSA, human-prostate-specific antigen; QDs, semiconductor quantum dots; GMP, guanine monophosphate; ThT@GMP/Eu, thioflavin T@GMP/Eu; CDs, carbon dots; AuNCs, gold nanoclusters.

## 3. Colorimetric Immunoassays

ALP-linked colorimetric immunoassays have shown great potential in clinical diagnosis due to their attractive advantages of cost-effectiveness and simple instrumentation, as well as facile and fast visual output [123]. The target concentration can be converted to the amount of ALP based on the interaction between the antigen and ALP-labeled antibody. Thus, the currently established colorimetric methods or chromogenic substrates for monitoring ALP activity can be directly adopted to the ALP-linked immunoassays [124]. Different colorimetric strategies, including the ALP-catalyzed production of chromogenic products, enzymatic-product-triggered chromogenic reaction, enzymatic-product-triggered plasmonic phenomenon, and enzymatic-product-mediated activity change of artificial enzymes or nanozymes, are systematically reviewed in this section (Figure 3).

### 3.1. ALP-Catalyzed Production of Chromogenic Product

ALP can catalyze the hydrolysis of colorless substrates to generate colored products (Figure 4). The qualitative detection of the target can be achieved through direct readout with the naked eye and by measuring the absorbance of the solution using a UV-vis spectrophotometer (Table 2) [125,126,127,128]. Based on this strategy, Darwish et al. reported an ELISA for 2-deoxycytidine detection based on ALP-catalyzed generation of PNP [129]. Jiang et al. developed a dual-colorimetric ELISA for the simultaneous determination of 13 fluoroquinolone and 22 sulfonamide residues in milk [130]. Although these immunoassays showed intensive color, the substrates are relatively toxic and the products are not stable. Thus, various novel strategies based on the enzymatic-product-mediated colorimetric reactions have continuously emerged.

### 3.2. Enzymatic-Product-Triggered Chromogenic Reaction

ALP-enzymatic products can react with specific reagents to cause a change in solution color [131]. For example, some metal complexes exhibit typical metal-to-ligand charge-transfer (MLCT) absorption properties, which can be modulated by the enzymatic products to produce visual and detectable signals [132]. Lei et al. reported a colorimetric immunoassay for rabbit IgG and PSA detection based on enzymatic-product-triggered in situ formation of purple-colored Cu(I)-bicinchoninic acid (BCA) complexes [133]. As shown in Figure 14A, ALP catalyzed the transformation of AAP into AA that could reduce Cu^2+^ into Cu^+^. The in situ formed Cu^+^-BCA complex exhibited a strong absorbance at 562 nm because of the LMCT absorption with a color change from light green to purple. In addition, the coordination polymer has been widely explored as a host to carry multiplex biomolecules with high loading efficiency for signal amplification in bioassays. Wu et al. developed a colorimetric immunoassay for the detection of CEA based on the antibody and ALP-loading ZnCPs and iron(II)-phenanthroline (Phen) complexes (Figure 14B) [134]. In this study, after the formation of the immunocomplex, the enzymatic product AA reduced Fe^3+^ into Fe^2+^ that could be coordinated with Phen to form the Fe^2+^-Phen complex with orange-red color. Finally, this cascade-amplified colorimetric method for CEA detection achieved a low LOD of 21.1 pg/mL. However, the low extinction coefficient of small molecules (both substrate and product) significantly limited the sensitivity of immunoassays.

Usually, the amount of ALP-enzymatic product to accelerate the colorimetric reaction is related to the amount of ALP labels and targets. Thus, trace targets only lead to the generation of a small quantity of color products and low level of signal change. Redox cycling has been proven to be a promising amplification strategy for the determination of trace targets in enzyme-linked biosensors [123]. Chen et al. reported a colorimetric immunoassay for AFP detection based on chemical redox cycling (Figure 15) [135]. In this work, the sandwich immune reaction was conducted in the polystyrene microplate. The ALP retained on the microplate catalyzed the hydrolysis of AAP into AA that could reduce colorless tris-(bathophenanthroline) iron(III) (Fe(BPT)_3_^3+^) encapsulated in the micelle of TX-100 into pink red tris(bathophenanthroline) iron(II) (Fe(BPT)_3_^2+^). Then, the formed oxidized product DHAA was reduced back into AA quickly in the presence of excess tris(2-carboxyethyl)-phosphine (TCEP), thus promoting the generation of abundant-color Fe(BPT)_3_^2+^ complexes. The LOD of the redox-cycling-based immunoassay was greatly improved (5 pg/mL), which is two orders of magnitude lower than that of conventional ELISA.

The change in the shade of the same color is always insensitive to the human eye. Thus, the multicolor transition with a ratiometric change in two constant-wavelength signals is more attractive in the development of colorimetric immunosensors [136]. Liu et al. developed an ALP-triggered ratiometric colorimetric and fluorescence immunosensor for the detection of fenitrothion using an NA-ALP fusion protein [137]. As illustrated in Figure 16, the substrate AAP could chelate with Fe^3+^ to produce an orange AAP-Fe^3+^ complex. Under ALP catalysis, AAP was transformed into AA, which showed no ability to chelate with Fe^3+^. The product AA could reduce potassium hexacyanoferrate(III) (K_3_[Fe(CN)_6_]) into potassium hexacyanoferrate(II) (K_4_[Fe(CN)_6_]) that subsequently reacted with Fe^3+^ to form Prussian blue [138]. Finally, the visible color transition from orange to blue was observed in the presence of fenitrothion. Meanwhile, AA could destroy MnO_2_ NSs into Mn^2+^ ions, thus inhibiting the oxidation of Na_2_SO_3_ by MnO_2_ NSs. The remaining Na_2_SO_3_ could convert 9-anthraldehyde into sulfuretted anthracene with a fluorescent color transition from green to blue. In addition, the ALP-enzymatic product can also inhibit the reactions between other reagents and chromogenic reaction substrates, which have been used to develop colorimetric immunoassays. For instance, Tang et al. demonstrated that Au(III) could oxidize 2,2′-azinobis(3-ethylbenzthiazoline-6-sulfonate) (ABTS) into a green product and the reduction of Au(III) by AA limited the reaction [139].

### 3.3. Enzymatic-Product-Triggered Plasmonic Phenomenon

Noble metal nanoparticles show fascinating size-, shape-, composition-, dielectric environment-, and distance-dependent LSPR properties and high extinction coefficients (e.g., 2.7 × 10^8^ M^−1^ cm^−1^ for AuNPs with an average diameter of 13 nm) [140]. The reducing ability of ALP substrates and products are different. For example, AAP is a non-reducing substance but ALP-catalyzed product AA can reduce metal ions such as Ag^+^ and Au^3+^ into colorful metal nanoparticles, causing the change of solution color. Thus, noble metal nanoparticles can serve as the ideal colorful chromogenic substrates for the development of ALP-mediated plasmonic colorimetric immunoassays.

#### 3.3.1. Enzymatic-Product-Induced Aggregation of Plasmonic NPs

ALP substrates and its enzymatic products can modulate the aggregation/dispersion states of plasmonic NPs with reversible color changes. Therefore, ALP can be coupled with plasmonic NPs to develop inter-nanoparticle distance-dependent colorimetric immunoassays [141,142]. The change in inter-nanoparticle distance will result in the shifted LSPR band and a pronounced color variance. Zhan et al. developed a dual-signal-amplified plasmonic immunoassay for the detection of respiratory syncytial virus based on the ALP-triggered dispersion of aggregated AuNPs [143]. As shown in Figure 17A, magnetic beads were used as the carriers to load thousands of ALP molecules for signal amplification. Zn^2+^ ions were added to enhance ALP activity by interacting with Pi ions (an inhibitor of ALP) to accelerate the dephosphorylation reaction of ATP. ATP with the negatively charged phosphate groups could induce the aggregation of positively charged cetyltrimethylammonium bromide (CTAB)-capped AuNPs via electrostatic interactions. The product adenosine from the ALP-catalyzed dephosphorylation of ATP made the AuNPs dispersed with the color change from gray to red.

ALP-triggered CuAAC reactions can cause the aggregation of alkyne- and azide-modified AuNPs [144]. Xianyu et al. developed an AuNP-based plasmonic colorimetric immunosensor for rabbit anti-human IgG detection based on ALP-triggered CuAAC chemistry [145]. As shown in Figure 17B, one ALP reporter could catalyze the production of numerous AA molecules. Then, the produced AA could reduce Cu^2+^ into Cu^+^ that further triggered the cycloaddition between alkyne- and azide-functionalized AuNPs, leading to the aggregation of AuNPs and a “red-to-blue” change in color. This method integrated the current immune detection platform with the AuNP-based plasmonic colorimetric assay, greatly enhancing the analytical performances.

**Figure 17 molecules-28-06565-f017:**
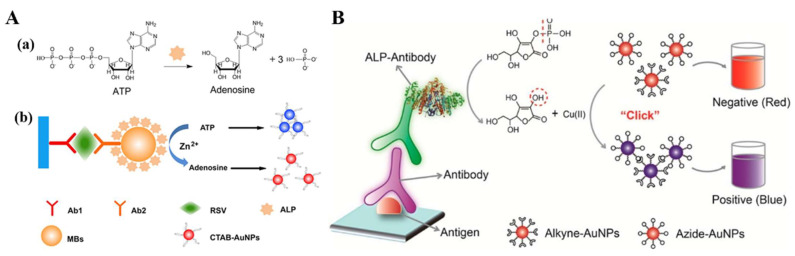
(**A**) Schematic illustration of the ALP-catalyzed dephosphorylation reaction for the sandwich plasmonic ELISA. (**a**) ALP catalyzes the dephosphorylation of ATP to generate adenosine and phosphate ions. (**b**) Dual-signal-amplified plasmonic ELISA based on the high loading of MBs and Zn^2+^-stimulated enzymatic reaction [143]. Copyright 2017 Elsevier. (**B**) Schematic illustration of naked-eye readout of plasmonic immunoassays based on ALP-triggered click chemistry [145]. Copyright 2014 American Chemical Society.

The surface modification of AuNPs is complicated. To address this issue and broaden the detection range, Ran et al. reported an ALP-linked colorimetric immunoassay for the simultaneous detection of multiple clinical biomarkers based on peptide-mediated cross-linking of modification-free AuNPs (Figure 18). In this approach, peptides with negatively charged phosphate groups near the alkaline groups on other residues could not induce the aggregation of negatively charged citrate-stabilized AuNPs. ALP could remove the negatively charged phosphate groups from the phosphorylated peptides through enzymatic dephosphorylation, thus causing the aggregation of AuNPs through the Au-N interactions. However, the above-mentioned aggregation-based colorimetric methods often suffer from false positive results due to the auto-aggregation induced by external factors, such as high ionic strength or other impurities in samples.

#### 3.3.2. Enzymatic-Product-Induced In Situ Metallization or Bioetching of Plasmonic NPs

The LSPR absorption properties of plasmonic NPs are closely related to several factors (e.g., shape, composition, size, and surrounding media), which can be readily regulated through ALP catalysis. Thus, many colorimetric plasmonic immunoassays have been developed on the basis of the ALP-triggered morphology change of pre-prepared NPs through in situ growth and bioetching methods. ALP can catalyze the conversion of inactive substrates into active reducing products that can induce the metallization on the pre-prepared metal NPs, leading to an obvious color change [147]. The enzymatic-product-regulated growth of AuNPs has been used to develop various kinds of colorimetric biosensors. Zhou et al. reported a colorimetric immunoassay for the detection of avian influenza virus based on ALP-enzymatic-product-induced silver metallization of AuNPs (Figure 19) [148]. In this study, PAPP was enzymatically hydrolyzed into PAP under the catalysis of ALP. The produced PAP served as a reducing agent to reduce Ag^+^ into Ag^0^ on the surface of AuNPs. The solution color changed from red to yellow, brown, or even black.

Unlike the absorption properties of spherical NPs, anisotropic gold nanostructures, such as gold nanorods (AuNRs), gold nanostars (AuNSs), and gold nanobipyramids (AuNBPs), exhibit characteristic transverse and longitudinal resonance plasmon absorption bands. The LSPR bands are more sensitive to aspect ratio, reshaping, composition, and other surrounding changes, which could be tuned from the visible to near-infrared region [149,150]. Wang et al. reported a multicolor immunosensor for visual detection of human epidermal growth factor receptor 2 (HER2 ECD) in serum based on AA-mediated in situ formation of AuNBPs with the assistance of reduced nicotinamide adenine dinucleotide I (NADH) [151]. As shown in Figure 20A, HER2 ECD was captured and then recognized by the ALP-labeled antibody in a sandwich format. NADH reduced the deep yellow Au(III) into colorless Au(I). In the presence of AAP, the enzymatic product AA could accelerate the growth of AuNBPs by reducing Au(I) into Au(0), producing a rainbow-like color change from colorless to wine red.

ALP-induced in situ metallization (generally Ag shell/coating) of anisotropic plasmonic metal nanomaterials can result in a significant change in the characteristic LSPR peaks of AuNRs, accompanied by a strong color change. This method can increase the signal-to-noise ratio of plasmonic immunoassays [155,156,157]. Yang et al. developed a colorimetric immunosensor for PSA detection based on the ALP-mediated reduction of Ag^+^ ions and deposition of ultrathin Ag shells on AuNRs [158]. Fu et al. reported a competitive colorimetric immunoassay for the detection of xanthylacrylamide (XAA) based on the monoclonal antibody (mAb) against XAA and the multicolor change of AuNRs [152]. As shown in Figure 20B, the mAb against XAA was labeled with ALP and used in the competitive immunoassay. ALP-enzymatic product AA reduced Ag^+^ into Ag shells on the AuNRs surface. The resulting strong gray-to-orange color change could be readily observed by the naked eye or a smartphone color detector. Because of the LSPR bands in the near-infrared optical window, AuNSs can absorb light and convert the energy into heat, resulting in a detectable temperature enhancement. Liu et al. developed a plasmonic and photothermal immunoassay for PSA detection through ALP-triggered in situ growth on AuNSs [153]. As displayed in Figure 20C, after the immunoreaction and ALP-catalyzed hydrolysis of AAP, the enzymatic product AA induced silver deposition on the surface of AuNSs. Due to the sensitivity toward the dielectric property of the surrounding medium, the LSPR of AuNSs showed a large blue shift in LSPR and a color change. Moreover, the photothermal conversion efficiency was also changed due to the shift in LSPR. As another example, Zha et al. reported a plasmonic and fluorescence immunoassay for the detection of acetochlor based on the ALP-triggered in situ growth of silver on AuNSs [159]. After the reduction of Ag^+^ by AA, the oxidized product of dehydrogenated AA could further interact with OPD, emitting a strong fluorescence.

In addition to anisotropic gold nanostructures, silver nanostructures can also serve as the substrates for plasmonic immunoassays. For instance, Kim et al. reported a plasmonic immunoassay for the detection of periodontal disease marker interleukin-1 beta (IL-1*β*) with the ALP-triggered geometrical transformation of Ag triangular nanoplates (AgNPLs) (Figure 20D) [154]. In this work, PAP reduced Ag^+^ into Ag seeds on the high-energy edge of AgNPLs, eventually resulting in the shape transformation from triangular to hexagonal, rounded pentagonal, and finally spherical. The blue shift in the LSPR absorption peak led to a multicolor response.

ALP-assisted bioetching strategies have been triumphantly introduced into the plasmonic immunoassays [160,161]. For example, Zhang et al. developed a plasmonic immunosensor based on ALP-triggered iodine-mediated etching of AuNRs [162]. As presented in Figure 21, after the formation of sandwich-type immunocomplexes, ALP catalyzed the hydrolysis of AAP into AA that could reduce I_2_, thus etching AuNRs from rod to sphere in shape and resulting in the blue-shift in the LSPR band and a color change from blue to red. This immunosensor achieved a naked-eye detectable LOD of 3 pg/mL for IgG detection.

### 3.4. Enzymatic-Product-Mediated Activity Change of Artificial Enzymes or Nanozymes

Molecular artificial enzymes and nanozymes have garnered a great deal of interest in bioassays because of their advantages of easy preparation, high stability, and low cost [163]. ALP can regulate the activities of molecular artificial enzymes by catalyzing their hydrolysis. Recently, Shi et al. developed a colorimetric immunosensor for AFP detection based on the ALP-controlled peroxidase-mimic activity of guanosine triphosphate (GTP) (Figure 22A) [164]. In this study, GTP with peroxidase-mimic activity could promote the H_2_O_2_-mediated oxidation of TMB. ALP catalyzed the dephosphorylation of GTP to guanosine 5′-diphosphate (GDP) and GMP, which both show no peroxidase-mimic activity toward the oxidation of TMB. AFP was sensitively determined with an LOD as low as 0.5 ng/mL.

ALP-catalyzed generation of active products can regulate the activity of nanozymes by controlling their generation/decomposition [165,166,167,168]. Zhang et al. developed a cascade-amplified immunoassay for OTA detection based on nanobody-ALP fusion and oxidase-like nanozyme MnO_2_ nanosheets, in which the enzymatic product AA could reduce MnO_2_ nanosheets into Mn^2+^ ions, limiting their oxidase-like activity toward TMB oxidation [169]. Lai et al. reported the colorimetric immunoassay of AFB_1_ based on the ALP-controlled in situ generation of nanozymes (Figure 22B) [170]. In this study, K_3_[Fe(CN)_6_] was reacted with Cu(II) to form copper hexacyanoferrate nanoparticles (CHNPs) that exhibited oxidase-mimicking activity. The reduction of K_3_[Fe(CN)_6_] into K_4_[Fe(CN)_6_] by AA inhibited the formation of CHNPs. In the ALP-mediated immunoassay with the aid of magnetic beads, ALP catalyzed the hydrolysis of AAP into AA that could inhibit the generation of CHNPs and the oxidation of the chromogenic substrate ABTS.

**Figure 22 molecules-28-06565-f022:**
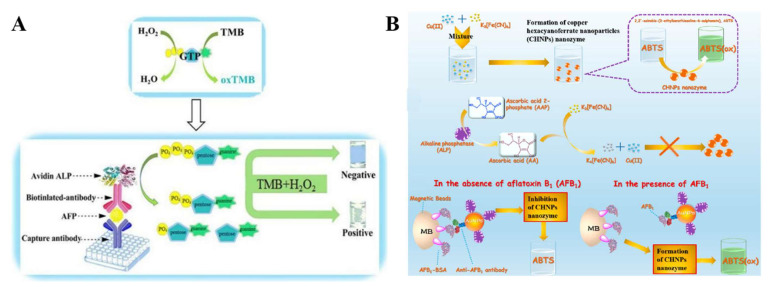
(**A**) Schematic illustration of AFP detection based on GTP-mediated enzyme cascade reaction [164]. Copyright 2018 Elsevier. (**B**) Schematic illustration of template-free just-in-time-producing copper hexacyanoferrate(III) strategy as oxidase-mimic-based colorimetric platform, enzyme-controllable-manner-based colorimetric platform, and CHNPs-ABTS-based colorimetric immunoassay [170]. Copyright 2022 Elsevier.

The in situ activation of nanozymes by ALP-enzymatic products is a novel cascade amplification strategy for colorimetric biosensing. As a proof, Jin et al. developed a colorimetric immunosensor for mouse IgG detection through enzymatic cascade reactions by combining ALP catalysis and the in situ generation of photoresponsive nanozymes [171]. As shown in Figure 23, during the immunoreactions, ALP catalyzed the hydrolysis of *o*-phosphoxyphenol (OPP) into catechol (CA) that could be further coordinated with TiO_2_ NPs via the specific and robust interaction between the enediol group of CA and Ti(IV) on the surface of TiO_2_. Under the illustration of visible light (*λ* ≥ 400 nm), the native TiO_2_ NPs were inert, but the formed CA-modified TiO_2_ NPs (TiO_2_-CA) exhibited oxidase-mimicking activity. The photoresponsive TiO_2_-CA nanozymes could catalyze the oxidation of TMB with dissolved O_2_ as the electron acceptor without the need for destructive H_2_O_2_. The LOD of this method was calculated to be 2.0 pg/mL for mouse IgG detection, which was 4500-fold lower than that of the traditional ELISA kit.

Although significant achievements have been witnessed in nanozyme-based fields, there are still some inherent drawbacks. First, the catalytic activity of most nanozymes is lower than that of natural enzymes. Second, the activity of nanozymes is closely related to the density of surface-active sites, and the modification of biomolecules on their surface for immunoassays may inhibit the catalytic activity. Enzyme-cascade-based multiple amplification strategies have been used in immunoassays. By taking advantage of the reducing activity of enzymatic products, ALP can be integrated with different natural enzymes, artificial enzymes, and nanozymes for signal amplification. For example, Xie et al. reported a colorimetric immunoassay for rabbit IgG detection by coupling enzymatic multicolor generation with a smartphone readout [172]. In this work, urease catalyzed the hydrolysis of urea, leading to a pH change that could be reflected by a color change in pH indicator phenol red from orange to red over 6.6 to 8.0. Ag^+^ could serve as the inhibitor to control the activity of urease. ALP-catalyzed product AA could reduce Ag^+^ into Ag, thus recovering the urease activity. Based on the ALP-adjusted urease-catalyzed multicolor generation system, rabbit IgG was detected with an LOD of 1.73 nm/mL.

ALP-based colorimetric immunoassays can be conducted using a plate reader or by the naked eye without the requirement of sophisticated instruments, providing an excellent alternative to achieve simple and rapid detection. The enzyme-regulated aggregation, growth, or bioetching of plasmonic Ag or Au nanoparticles and the cascade reactions between ALP and nanozyme catalysis can greatly enhance the detection sensitivity. However, there are still several problems, such as low accuracy, unstable chromogenic substrates, and the interfering effect from the colored matrix in biological samples. Efforts should be devoted to the combination of ALP and chromogenic substrates and nanomaterials to improve their stability in harsh environmental conditions.

**Table 2 molecules-28-06565-t002:** Analytical performance of ALP-based colorimetric immunoassays.

Detection Principle	ALP Substrates	Chromogenic Substrates/Reactions	Target	Linear Range	LOD	Reference
ALP-catalyzed production of chromogenic product	PNPP	PNPP	IgG	0.5–400 ng/mL	62.5 pg/mL	[126]
	PNPP	PNPP	TNF-α	0–10 ng/mL	120 pg/mL	[127]
	3-IP	3-IP	Mouse IgG	0.3–250 ng/mL	0.3 ng/mL	[128]
	PNPP	PNPP	2-Deoxycytidine	10–1000 μM	Not reported	[129]
Enzymatic-product-triggered chromogenic reaction	PAPP	The reaction between diethanolamine and PAP	AFP	0.1–20 ng/mL	0.1 ng/mL	[131]
	AAP	Cu(I)-bicinchoninic complex	Rabbit IgG	0.1–25 ng/mL	0.05 ng/mL	[133]
	AAP	Fe(III)-phenanthroline complex	CEA	0.05–100 ng/mL	21.1 pg/mL	[134]
	AAP	Fe(III)- tris-(bathophenanthroline) complex	AFP	0.01–5 ng/mL	5 pg/mL	[135]
	APP	In situ formation of Prussian blue	PSA	1–800 ng/mL	1.2 ng/mL	[136]
	APP	In situ formation of Prussian blue	Fenitrothion	4.7–11.6 ng/mL	3 ng/mL	[137]
Enzymatic-product-induced aggregation of plasmonic NPs	AAP	Mn^2+^-mediated aggregation of AuNPs	Fumonisin B1	6.25–200 ng/mL	0.15 ng/mL	[142]
	ATP	Zn^2+^-mediated aggregation of AuNPs	Respiratory syncytial virus	0.1–30 pg/mL	21 fg/mL	[143]
	AAP	AuNPs-based click reaction	Norfloxacin	3.18 × 10^−2^–6.88 × 10^3^ pg/mL	10fg/mL	[144]
	Peptide	AuNPs	PCT, IL-6, CRP	0.2–25 ng/mL, 50–1600 pg/mL, 3.15–100 μg/mL	0.24 ng/mL, 12.5 pg/mL, 1.15 μg/mL	[146]
Enzymatic-product-induced in situ metallization or bioetching of plasmonic NPs	AAP	Ag growth on SiO_2_@AuNPs	IgG	0.7–70 pM	0.14 pM	[147]
	PAPP	Ag growth on AuNPs	H9N2 AIV	0.02–1 ng/mL	17.5 pg/mL	[148]
	AAP	Growth of AuNPs	Tyramine	0.313–20 mg/L	0.246 mg/L	[149]
	AAP	Growth of AuNPs	HER2 ECD	1–7 ng/mL	0.05 ng/mL	[151]
	AAP	Ag growth on AuNRs	Xanthylacrylamide	0.3–17.2 ng/mL	0.06 ng/mL	[152]
	AAP	Iodine-mediated etching of AuNRs	Human IgG	0.1–10 ng/mL	100 pg/mL	[162]
Enzymatic-product-mediated activity change of artificial enzymes or nanozymes	GTP	GTP-accelerated TMB oxidation	AFP	1–100 ng/mL	0.5 ng/mL	[164]
	AAP	In situ generated CHNPs to catalyze ABTS oxidation	Aflatoxin B_1_	1 pg/mL–20 ng/mL	0.73 pg/mL	[170]
	AAP	In situ generated PdNPs to catalyze TMB oxidation	PSA	5–50 ng/mL	1 ng/mL	[168]

Abbreviation: PNPP, *p*-nitrophenyl phosphate; IgG, immunoglobulin G; TNF-α, tumor necrosis factor α; 3-IP, 5-bromo-4-chloro-3-indolyl phosphate; PAPP, *p*-aminophenyl phosphate; PAP, *p*-aminophenol; AFP, α-fetoprotein; AAP, L-ascorbic acid 2-phosphate trisodium salt; CEA, carcinoembryonic antigen; PSA, human-prostate-specific antigen; AuNPs, gold nanoparticles; ATP, adenosine triphosphate; PCT, procalcitonin; IL-6, interleukin-6; CRP, C-reactive protein; PAPP, *p*-aminophenyl phosphate monohydrate; AIV, avian influenza virus; HER2 ECD, extracellular domain of human epidermal growth factor receptor 2; AuNRs, gold nanorods; GTP, guanosine triphosphate; CHNPs, copper hexacyanoferrate nanoparticles; ABTS, 2,2′-azinobis(3-ethylbenzthiazoline-6-sulfonate); TMB, 3,3′,5,5′-tetramethylbenzidine.

## 4. Chemiluminescence Immunoassays

Chemiluminescence systems can be combined with enzyme-linked immunoassays to detect various targets, in which the chemiluminescence signals are changed through chemical or biochemical reactions. In ALP-linked chemiluminescence immunoassays, ALP catalyzes the hydrolysis of substrates into chemiluminescence reagents or the products that can react with other reagents to emit luminescence (Figure 5) [173,174,175,176,177]. For example, Nie et al. developed a chemiluminescence immunoassay using an optical-fiber sensor for all-directional signal collection with high efficiency, in which the amounts of ALP and target cTnI were determined by the commercial chemiluminescence reagent APS-5 [178]. Zhao et al. developed a chemiluminescence immunoassay for total PSA and free PSA detection based on the reactions between HRP and luminol as well as ALP and 3-(2′-spiroadamantyl)-4-methoxy-4-(3″-phosphoryloxy)-phenyl-1,2-dioxetane (AMPPD) [179]. In addition, Lucigenin (*N*,*N*′-dimethyl-9,9′-biacridinium substrate) can react with organic reductants (e.g., AA, NADH, and phenacyl alcohol) in alkaline medium, which can be produced through the ALP-catalyzed substrate hydrolysis [180,181]. ALP and HRP can simultaneously serve as the label to catalyze the chemiluminescence reaction for multiplex measurement of targets [182,183]. Hu et al. developed a chemiluminescence immunosensor through bioorthogonal reaction for the multi-detection of hepatocellular carcinoma biomarkers (Figure 24A) [184]. In this study, carbon nanotubes (CNTs) were modified with HRP, ALP, and antibodies for AFP and Golgi protein 73. After the immunoreaction, HRP and ALP catalyzed the corresponding substrates to emit chemiluminescence that could be recorded in different time windows.

In the presence of ATP, luciferase can catalyze the oxidation of ofluciferin to oxyluciferin with bioluminescence. Therefore, ALP-linked immunoassays can be developed based on the ATP−luciferin−luciferase bioluminescent reaction through the ALP-catalyzed hydrolysis of ATP. For instance, Chen developed an ALP/luciferase double-enzyme-mediated bioluminescent biosensor for the detection of procalcitonin (PCT) [185]. As shown in Figure 24B, after the magnetic immunoreaction, ALP catalyzed the dephosphorylation of ATP and hence hindered the ATP-luciferin-luciferase bioluminescent reaction. The decreased bioluminescence intensity was monitored using a portable ATP detector.

The combination of ALP catalysis with chemiluminescence technology endows immunoassays with a high sensitivity, fast response, and wide dynamic range. Several considerable challenges in practical applications of ALP-linked chemiluminescence immunoassays need to be addressed. Novel nanomaterials should be synthesized to improve the stability of chemiluminescence enhancers and amplify the signal intensity. Other technologies can be coupled to ALP-based chemiluminescence immunoassays for microminiaturization, automation, and cost-effective and multiplex analysis, such as flow injection, optical fibers, (on-chip) capillary electrophoresis, and microfluidic technology.

## 5. SERS Immunoassays

Compared with the conventional Raman scattering, SERS can enhance the Raman scattering intensity of molecules by up to 10~14 orders of magnitude when Raman molecules are adsorbed on the rough solid metal or plasmonic nanostructure surface [186,187]. It can provide narrow and well-resolved spectroscopic bands and highly improve the detection sensitivity. Therefore, SERS techniques can be combined with ALP-linked immunoassays by measuring the enzymatic products or monitoring the ALP-catalysis-induced changes in Raman signals [188,189].

ALP can catalyze the transformation of SERS-inactive substrates into SERS-active species. The enzymatic products can generate molecularly specific Raman signals by adsorbing on the surface of the SERS interface [190]. Campbell et al. developed an ALP-labeled SERS immunoassay for the detection of C-reactive protein (CRP) using bromochloroindolylphosphate as the substrate [191]. ALP catalysis on the substrate resulted in the formation of SERS-active insoluble dimers with a distinctive peak at 600 cm^−1^. In addition, Chen et al. fabricated an AgNPs/polymer/filter paper SERS substrate to determine the amount of insoluble ALP products [192].

The employment of Ag and Au NPs as the plasmonic substrates can enhance the Raman signals. Thus, signal amplification strategies in ALP-mediated plasmonic immunoassays can be introduced into SERS platforms, such as the AA-triggered click reaction and enzymatic-product-induced aggregation of AuNPs [193]. For example, ALP-catalyzed silver metallization on the substrate or AuNPs can enhance the Raman signal of dyes. Chen et al. reported an SERS immunoassay for human IgG detection based on the ALP-catalyzed production of AgNPs to enhance the signal of Raman dyes [194]. Yang et al. reported an SERS immunoassay for AFP detection based on ALP-triggered Ag metallization on AuNPs, in which the in situ formation of Ag layer increased the Raman signal of 4-mercaptobenzoic acid modified on AuNPs [195]. In addition, nucleic acid amplification technology is a powerful strategy for ultrasensitive bioassays and the formed DNA structures can be further labeled with numerous ALP molecules for secondary signal amplification [196]. Wang et al. reported a heterosandwich SERS immunoassay for total PSA (tPSA) detection based on ALP-induced silver deposition on AuNRs and DNA nanofirecrackers formed through HCR [197]. As shown in Figure 25, the PSA aptamer was conjugated with the initiator primer and the antibody-antigen-aptamer heterosandwich structures were formed on the plate in the presence of tPSA. The initiator primer retained on the plate triggered the bilayer HCR process to generate hyperbranched DNA nanostructures with multiple biotin groups. Many SA-labeled ALP conjugates were then linked to the DNA nanostructures, effectively catalyzing the hydrolysis of AAP. The resulting AA could reduce Ag^+^ into Ag shells on AuNRs, leading to the significant enhancement of the Raman signal.

ALP-linked SERS immunoassays show excellent performances in terms of sensitivity, selectivity, and multiplexing capability. Nevertheless, there are still many challenges associated with the development of such immunoassays. For example, novel SERS tags and uniform SERS substrates with high sensitivity and reproducibility and low cost are required, and the “Raman spectrofingerprint” databases of various antigens are desired.

## 6. Conclusions

ALP, as a reporter enzyme in immunoassays, holds significant importance in amplifying detectable signals arising from immune-recognition events. Although the classic ALP-linked optical immunoassays are simple and effective, they inherently exhibit low sensitivity and high background noise. The abundance of substrate/product pairs and the efficient ALP catalysis reactions offer promising ways to seamlessly couple conventional immunoassays with emerging powerful strategies. For example, colorimetric assays provide facile, simple, and fast detection due to their vivid visual signal, while fluorometric methods show high selectivity and low LOD due to their self-calibration capability. To enhance sensitivity and selectivity, various signal amplification strategies have been integrated with ALP-linked immunoassays, including the utilization of nanomaterials as carriers to load more enzymes, DNA-based amplification techniques, enzymatic cascade reactions, in situ generation of nanozymes, and multicolor plasmonic metal NPs. In particular, ALP enzymatic products can modulate the physicochemical properties of nanomaterials, inducing significant changes in the optical signals of nanomaterials. Meanwhile, highly sensitive optical immunoassays can be designed by combining ALP and nanozyme catalysis.

Despite notable success, several challenges still persist in the domain of ALP-linked optical immunoassays. First, most of the proposed amplification strategies rely on the reductive or coordinative ability of ALP-catalyzed products to regulate the optical or catalytic properties of nanomaterials. However, reducing reagents in complex samples may intervene in the corresponding reactions in washing-free immunoassays. Thus, more efforts should be focused on the exploration of effective approaches to decrease the interference-mediated false signals. Second, ALP substrates with interior stabilities can limit the applicability of ALP-linked immunoassays and the enzymatic product of AA in the fluid matrix can cause a false result. Therefore, newly developed ALP substrates with excellent stability under various physiological conditions are still required in actual sample analysis. Third, the existing methods for enzyme/antibody immobilization on nanomaterials may encounter issues such as enzyme leaching, denaturation, complex procedures, and decreased recognition ability. When nanomaterials are used as the nanocarriers or substrates, their size, shape, or composition may obviously affect the reproducibility and accuracy of measurement results. Therefore, effective approaches that can improve the quality of nanomaterials and the bioconjugation efficiency are attractive in future. Fourth, the POCT system has gained immense popularity due to its portability and cost-effectiveness. It is imperative to invest more efforts into integrating ALP-linked immunoassays with smartphones, wireless communication, cloud computing, and data storage technologies for simultaneous multiplex analysis of different biomarkers. In short, the strategic incorporation of ALP in immunoassays has proven to be instrumental in amplifying detectable signals and achieving high sensitivity and selectivity. Advancements in nanomaterial-based approaches, multiplex analysis, and POCT systems present exciting opportunities for overcoming existing challenges and expanding the applicability of ALP-linked immunoassays in various domains, including clinical diagnostics and environmental monitoring.

## Data Availability

Not applicable.

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
