# Peer review of "Overview on the Development of Alkaline-Phosphatase-Linked Optical Immunoassays"

_molecules, 2023, doi:10.3390/molecules28186565_

Round 1

Reviewer 1 Report

The present paper summarize a lot of literature concerning ALP application for development of different biosensing platforms. The manuscript is well written. Authors categorized all sensors according to the type of analytical signal. The review is well illustrated and provide all the needed references. 

I believe that it would be beneficial for authors to add a table summarizing all the reviewed biosensors in term of detected targets, type of analytical signal, LOD, selectivity and specificity. It would be very convenient for the future reader to have all this information in one place. It would be also convinient if authors provide a chemical structures for ALP substrates in the Introduction section and compare them.

The manuscript needs minor English language editing.

Author Response

We thank the reviewer for his/her positive and constructive comments: The present paper summarize a lot of literature concerning ALP application for development of different biosensing platforms. The manuscript is well written. Authors categorized all sensors according to the type of analytical signal. The review is well illustrated and provide all the needed references.

Comment 1: I believe that it would be beneficial for authors to add a table summarizing all the reviewed biosensors in term of detected targets, type of analytical signal, LOD, selectivity and specificity. It would be very convenient for the future reader to have all this information in one place. It would be also convenient if authors provide a chemical structures for ALP substrates in the Introduction section and compare them.

Response: It is a good suggestion. We have added two tables to compare the analytical performances of ALP-linked optical immunoassays. We also added two schemes to present the chemical structures of ALP substrates.

Comment 2: The manuscript needs minor English language editing.

Response: We have asked as an English speaker to revise the language carefully.

Reviewer 2 Report

This manuscript adresses an interesting and relevant topic. 

Please allow some general and some more specific comments:

1. I was a little bit surprised not to read more about digital immunoassays, such as https://pubs.rsc.org/en/content/articlelanding/2015/AN/C5AN00714C

2. Many figures seem to be too small. For example in Figure 1, A and B should be below each other. Fig. 15 should be simply larger to use the width of the page.

3. In some figures, in the attempt to show the general concepts, for me it was difficult to understand the mechanism on a molecular level. Perhaps it would be nice to have a figure showing all known catalytic reactions of ALP.

4. At the end of the paper, I missed a more detailed comparison. The abstract starts with "The drive for achieving ultrasensitive target detection ..." but these data (LOD and so on), are not shown. A comparison table with the different approaches would be a big plus. 

5. The conclusions are quite general and could be written in a similar way for many different approaches and labels. A more specific and even personal opinion might be given, on why ALP should be used (and not other labels).

Author Response

We thank the reviewer for his/her positive comments: This manuscript addresses an interesting and relevant topic. Please allow some general and some more specific comments:

Comment 1: I was a little bit surprised not to read more about digital immunoassays, such as https://pubs.rsc.org/en/content/articlelanding/2015/AN/C5AN00714C

Response: Digital assays have been used for the detection of ALP but few ALP-linked digital immunoassays have been reported. We have discussed the works of digital immunoassays and cited the references.

Comment 2: Many figures seem to be too small. For example in Figure 1, A and B should be below each other. Fig. 15 should be simply larger to use the width of the page.

Response: We have improved the quality of figures.

Comment 3: In some figures, in the attempt to show the general concepts, for me it was difficult to understand the mechanism on a molecular level. Perhaps it would be nice to have a figure showing all known of ALP.

Response: It is a good suggestion. We have added two schemes to show the chemical structures of ALP substrates and the catalytic reactions.

Comment 4: At the end of the paper, I missed a more detailed comparison. The abstract starts with "The drive for achieving ultrasensitive target detection ..." but these data (LOD and so on), are not shown. A comparison table with the different approaches would be a big plus.

Response: We have added two tables to compare the analytical performances of the ALP-linked optical immunoassays.

Comment 5: The conclusions are quite general and could be written in a similar way for many different approaches and labels. A more specific and even personal opinion might be given, on why ALP should be used (and not other labels).

Response: We have revised the conclusion carefully and discussed the merits of ALP as signal labels in Introduction.

Round 2

Reviewer 2 Report

I am still not satisfied with this manuscript. In fact, some additional problems have arisen:

The comment that AP has a 1000x higher turnover number than HRP seems strange to me. According to Porstmann (1985), HRP is faster. In addition, it is not mentioned that HRP is inhibited by its own substrate  H2O2. 

The shown substrates seem to be quite arbitrary selection, perhaps some are even wrong. The very well-known chemiluminescence substrate AMPPD is also not mentioned. As far as I know, Nitrophenylphosphate is the by far most popular substrate. Hence, this should be shown first.

I find it strange to say that in AP reactions, reductions would occur. From my knowledge, AP is a hydrolase and not a redox enzyme.

The issue with the small figures has not been resolved throughout. Most of the figures were unchanged. In addition, most figures should be delivered in a higher resolution as a PNG or TIFF, and not as a JPG.

In line 242, a typo seems to have occurred: "CHzyme-catalyzed reaction"

In Table 1, it would be more convenient to show the linear range and the LODs in the same unit (in the table head). In addition, chromogenic and chemiluminescent substrates are lacking here. 

For me, a thorough discussion is still lacking. Which of the many variants is the best and why?

Author Response

Comment 1: The comment that AP has a 1000x higher turnover number than HRP seems strange to me. According to Porstmann (1985), HRP is faster. In addition, it is not mentioned that HRP is inhibited by its own substrate  H2O2.

Response: This statement is based on the reference literature (Analyst, 2014, 139, 439–445). We have changed the sentence into "ALP has garnered considerable attention as a reporter enzyme for signal amplification due to its outstanding advantages of high catalytic activity, high turnover number, and broad substrate specificity ".

Comment 2: The shown substrates seem to be quite arbitrary selection, perhaps some are even wrong. The very well-known chemiluminescence substrate AMPPD is also not mentioned. As far as I know, Nitrophenylphosphate is the by far most popular substrate. Hence, this should be shown first.

Response: The generated substrates for fluorescence, colorimetry and chemiluminescence assays were shown in Schemes 2, 4, and 5. We have revised the schemes carefully.

Comment 3: I find it strange to say that in AP reactions, reductions would occur. From my knowledge, AP is a hydrolase and not a redox enzyme.

Response: We are sorry for this mistake. The reduction reactions occur between metal ions and enzymatic products such as AA but not ALP. We have revised the statement and checked the manuscript carefully.

Comment 4: The issue with the small figures has not been resolved throughout. Most of the figures were unchanged. In addition, most figures should be delivered in a higher resolution as a PNG or TIFF, and not as a JPG.

Response: The re-used figures are provided by the authors with the permission of publishers. We have improved the figure resolution ASAP.

Comment 5: In line 242, a typo seems to have occurred: "CHzyme-catalyzed reaction"

Response: We have defined the reaction in the revised manuscript.

Comment 6: In Table 1, it would be more convenient to show the linear range and the LODs in the same unit (in the table head). In addition, chromogenic and chemiluminescent substrates are lacking here.

Response: We added the chromogenic and chemiluminescent substrates in Scheme 4 and 5. Different targets including proteins, enzymes, cells, bacteria and exosomes have been determined by ALP-linked fluorescence immunoassays. The linear range and LODs are not shown in the same unit since the units of different targets are distinctive.

Comment 7: For me, a thorough discussion is still lacking. Which of the many variants is the best and why?

Response: We thank the reviewer for his/her comments. Each method has its advantages and disadvantages. We have added the comments on fluorescence, colorimetry and chemiluminescence assays at the end of each part and revised the conclusion carefully.

Addressing the reviewer’s comments has improved the quality of our work. Thank you very much again for your consideration of publication of our manuscript in Molecules.